# Association between *Interferon-Lambda-3* rs12979860, *TLL1* rs17047200 and *DDR1* rs4618569 Variant Polymorphisms with the Course and Outcome of SARS-CoV-2 Patients

**DOI:** 10.3390/genes12060830

**Published:** 2021-05-28

**Authors:** Sara H. A. Agwa, Marwa Mostafa Kamel, Hesham Elghazaly, Aya M. Abd Elsamee, Hala Hafez, Samia Abdo Girgis, Hoda Ezz Elarab, Fatma S. E. Ebeid, Safa Matbouly Sayed, Lina Sherif, Marwa Matboli

**Affiliations:** 1Molecular Genomics Unit, Clinical Pathology Department, Medical Ain Shams Research Institute (MASRI), School of Medicine, Ain Shams University, Cairo 11566, Egypt; 2Medicinal Biochemistry and Molecular Biology Department, School of Medicine, Ain Shams University, Cairo 11566, Egypt; Marwa.kamel88@med.asu.edu.eg; 3Oncology Department, Medical Ain Shams Research Institute (MASRI), Ain Shams University, Cairo 11566, Egypt; heshamelghazaly@hotmail.com; 4Molecular Genomics Unit, Medical Ain Shams Research Institute (MASRI), Ain Shams University, Cairo 11566, Egypt; aya_ana2025@yahoo.com; 5Infection Control Unit, Clinical Pathology Department, Ain Shams University Hospitals, Cairo 11566, Egypt; Dr.hmhafez@gmail.com (H.H.); Drsamia.girgis@med.asu.edu.eg (S.A.G.); Hodaezz55@yahoo.com (H.E.E.); 6Pediatric Department, School of Medicine, Ain Shams University Hospitals, Cairo 11566, Egypt; fatmaebeid@med.asu.edu.eg (F.S.E.E.); safamatboly@yahoo.com (S.M.S.); 7Department of Clinical Pharmacy, Faculty of Pharmacy, Misr International University, Cairo 11566, Egypt; Lina1405218@miuegypt.edu.eg

**Keywords:** severe acute respiratory syndrome coronavirus 2, coronavirus, immune response, innate immunity, polymorphisms, single nucleotide, cytokine

## Abstract

Background: Severe acute respiratory syndrome coronavirus 2 (SARS-CoV-2) infection provides a critical host-immunological challenge. Aim: We explore the effect of host-genetic variation in interferon-lambda-3 rs12979860, Tolloid Like–1 (*TLL1*) rs17047200 and Discoidin domain receptor 1(*DDR1*) rs4618569 on host response to respiratory viral infections and disease severity that may probe the mechanistic approach of allelic variation in virus-induced inflammatory responses. Methods: 141 COVID-19 positive patients and 100 healthy controls were tested for interferon-lambda-3 rs12979860, *TLL1* rs17047200 and DDR1 rs4618569 polymorphism by TaqMan probe-based genotyping. Different genotypes were assessed regarding the COVID-19 severity and prognosis. Results: There were statistically significant differences between the studied cases and control group with regard to the presence of comorbidities, total leucocytic count, lymphocytic count, CRP, serum LDH, ferritin and D-dimer (*p* < 0.01). The CC genotype of rs12979860 cytokine, the AA genotype of *TLL1* rs17047200 and the AA genotype of the rs4618569 variant of *DDR1* showed a higher incidence of COVID-19 compared to the others. There were significant differences between the rs4618569 variant of *DDR* and the outcome of the disease, with the highest mortality in AG genotype 29 (60.4%) in comparison to 16 (33.3%) and 3 (6.2%) in the AA and GG genotypes, respectively (*p* = 0.007*), suggesting that the A allele is associated with a poor outcome in the disease. Conclusion: Among people who carry C and A alleles of SNPs *IFN-λ* rs12979860 and *TLL1* rs17047200, respectively, the AG genotype of the DDR1 rs4618569 variant is correlated with a COVID-19 poor outcome. In those patients, the use of anti-*IFN-λ 3*, *TLL1* and *DDR1* therapy may be promising for personalized translational clinical practice.

## 1. Introduction

Most cases with severe acute respiratory syndrome coronavirus 2 (The SARS-CoV-2) infection demonstrate mild symptoms or are asymptomatic, but some cases may develop complications such as interstitial pneumonia and acute respiratory distress syndrome (ARDS), as seen in patients with advanced age and associated morbidities [1].

The relation between SNPs and the immune response is called “immunogenetic profiling” [2]. Genetic polymorphism plays an important role in cytokine function, which is crucial in the host inflammatory response [3]. Cytokine polymorphisms affect gene transcription and expression, and subsequently the amount of cytokine secreted according to the genotype of the cytokine [4,5]. The cytokine polymorphisms could be associated with rheumatoid arthritis, ankylosing spondylitis and systemic lupus erythematosus [6].

Knowing that cytokines are crucial regulators of the individual response to infections [7], it is becoming clear that the immune system has an evident role in the cytokine storm as dangerous events in the disease known as ARDS [8,9]. Understanding the variables between subjects that could lead to this outcome is the cornerstone for identifying targeted therapeutic strategies [10]. 

rs12979860 is a single nucleotide polymorphism (SNP) previously known as interferon-lambda-3 (IFN-lambda-3). Retrieved from literature, it was found to be connected with the host viral immune defense [11,12,13]. The rs12979860 SNP is found upstream of the *IFN-λ 3* gene in the promoter sequence. It produces *IL28A* (*IFN-λ 2*), *IL28B* (*IFN-λ 3*) and *IL29* (*IFN- λ 1*), members of cytokines in the *IFN- λ* family [14]. They share in the regulation of the host immune response in viral infections [15]. 

*TLL1* rs17047200 SNP may produce catalytically highly active short isoform (*TLL1* isoform 2) [16]. Of note, we evaluated a SNP of Tolloid Like–1 (*TLL–1*) as a complementary activating protease and also as potentially able to stimulate the spike protein of SARS-CoV-2 [17].

Discoidin domain receptor 1 (*DDR1*) is a tyrosine kinase receptor that plays a role in b1 integrins signaling [18]. *DDR1* activation by collagen affects human leukocytes’ functions as cell differentiation and cytokine production [19]. *DDR1* modulates E-cadherin and integrin CCN3-dependent cell adhesion to collagen type IV [20]. We focused on cytokine polymorphisms at rs12979860 and rs17047200 loci, the allele frequencies reported in previous literatures and *DDR1* gene polymorphism in COVID-19 positive patients to explore their relation to coronavirus infection severity and mortality.

The goal of the study is to assess the association between rs12979860, *TLL1* rs17047200 and *DDR1* rs4618569 polymorphism and coronavirus disease of 2019 (COVID-19) outcomes, because the earlier prediction of the host genotype and the severity of the disease will alleviate the economic burden or mortality rate of COVID-19.

## 2. Materials and Methods

### 2.1. Study Design and Demographic Data

This is a case-control study that was performed on blood samples from patients that were admitted to the pediatric, ENT, chest and internal medicine department at Ain Shams university hospitals.

We recruited 141 COVID-19 positive patients, including 23 (16%) from pediatric patients, 87 (61.7%) from the chest department and 31 (22%) from the geriatric department, and 100 healthy controls coming for a routine check-up visit during the period from May 2020 to March 2021. COVID-19 diagnosis depends on the characteristic clinical and computed tomography (CT) scan of the chest and confirmed by the positive qRT-PCR for COVID-19 in a nasopharyngeal swab. COVID-19 was classified as 82 mild cases (58%), 11 severe cases (7.8%) and 48 critical cases (34%). Classification of COVID-19 patients’ severity was according to the Egyptian MOH protocol, version 1.4 [21] into mild, moderate, severe or critical groups. Patients with mild cases are either asymptomatic or display leucopenia or lymphopenia and without pneumonia in a CT image. Severe cases had a respiratory rate > 30 breaths/min or a PaO_2_/FiO_2_ ratio < 300, SpO_2_ ≤ 92% in room air or a CT showing higher than 50% progressive lesion within 1–2 days. Critical cases had a PaO_2_/FiO_2_ ratio < 300 despite O2 therapy or SpO_2_ ≤ 92% in room air or a respiratory rate >30 breaths/min.

A complete history was taken from each patient focusing on (i) history of comorbidities as diabetes, hypertension, asthma or combined; (ii) severity of the disease (whether ventilated or not). A complete blood count—D-dimer, serum ferritin, C-reactive protein (CRP), lactate dehydrogenase (LDH)—was performed on all participants at the time of diagnosis of SARS-CoV-2 infection.

Patient nasopharyngeal swabs were collected for viral RNA isolation followed by purification using the QIAamp Viral RNA Mini kit (Cat no. 52906; Qiagen, Tokyo, Japan) based on the manufacturer’s guidelines.

### 2.2. SNP Genotyping

#### 2.2.1. Extraction of Total RNA from Whole Blood Samples

SNPs genotyping was carried out on the extracted RNA. We used the QIAamp RNA Blood Mini kit (Qiagen, Valencia, CA, USA) to purify total RNA from the whole blood samples based on the manufacturer’s guidelines. We assessed the RNA concentration and integrity using the Qubit 3.0fluorometer (Invitrogen, serial no. 2321609092). Then, the extracted total RNA was reverse-transcribed into cDNA by the high-capacity cDNA Reverse Transcription kit (A. B., Thermo Fisher Scientific, Waltham, MA, USA) according to the manufacturer’s protocol on the Thermo Hybaid polymerase chain reaction PCR express (Thermo Scientific, Missouri City, TX, USA).

#### 2.2.2. Genotyping Methodology

The SNP assay coded by C_33773674_10 (rs17047200) Lot: p180806- 009 C03, C _7820464_10 (rs12979860) Lot: p 180115- 001 E12 and C_ 2437004_20 (rs4618569) Lot: P 200101- 000 B06 Applied Biosystems predesigned TaqMan SNP genotyping Assays (cat.no: 4351379, Thermo Scientific, A.B., Missouri City, TX, USA) and TaqMan universal master mix II (PN: 4428173, Thermo Scientific, A.B., Missouri City, TX, USA) were genotyped using the TaqMan genotyping allelic discrimination method. All PCR reactions were done in 50 μL reaction volumes: 25 μL TaqMan Universal Master Mix II, 2.5 μL TaqMan Assay and cDNA+RNase-free water 22.5 μL to reach a total final volume of 50 μL per reaction. PCR reactions were performed on the Applied Biosystems Tm 7500 Real-Time PCR system (Foster City, CA, USA). The following thermocycler conditions were adopted: 95 °C for 10 min hold and denaturation at 95 °C for 15 s, then annealing and extension at 60 °C for 1 min in 40 cycles.

Endogenous control genes were used to normalize the raw data of the samples followed by comparing the results to a reference sample. In this study, MIQE guidelines were followed to avoid any experimental error during extraction and RNA processing.

#### 2.2.3. Statistical Analysis

Statistical social packages of SPSS software version 20 were used. Qualitative data were described using the number and percent. Quantitative data were analyzed using the median for non-parametric data and the mean ± SD for parametric data. The statistical significance was considered at the (0.05) level. Qualitative data were analyzed using the Chi-Square test. The statistical relevance of the relationship between genotypes and risk of COVID-19 was expressed by an odds ratio (OR) with its 95% confidence interval (95% CI). The Hardy–Weinberg equilibrium was applied to compare the genotype distribution among the studied groups. Spearman Rank Correlation tests were assessed between different genotypes. 

## 3. Results

### 3.1. Patient Sociodemographic and Clinicopathological Features

No statistically significant differences were found between COVID-19 positive patients and controls regarding age or sex with *p* = 0.252 and 0.201, respectively. The mean age of the patients was 37.42 ± 20.51, and females represent 43.2% of cases. Comorbidities (including diabetes mellitus, hypertension, chronic pulmonary disease, renal disease and cardiac disease) were found in 57.4% of cases and 18% of controls. There was a significant difference between the COVID-19 patients and the healthy control group with regard to the presence of comorbidities, total leucocytic count, lymphocytic count, CRP, serum LDH, ferritin and D-dimer (*p* < 0.01) (Table 1).

### 3.2. Genotypes Characteristics

TaqMan-based genotyping revealed the following: With regard to the interferon-lambda-3 rs12979860, 19 (13.5%) patients were TT, 59 (41.8%) were TC and 63 (44.7%) were CC. With regard to the control group, 12 (12%) of them were TT, 44 (44%) were TC and 44 (44%) were CC using χ^2^: Chi-Square test (Table 2). Remarkably, it seems that the CC genotype of the rs12979860 cytokine showed a higher incidence of COVID-19 compared to the healthy control group (*p* = 0.011).

With regard to TLL1 rs17047200, 94 (66.7%) patients were AA, 37 (26.2%) were AT and 10 (7.1%) were TT. The healthy control group was represented as 66 (66%) AA, 26 (26%) AT and 8 (8%) TT (Table 2). From these data, it appears that the AA genotype of the TLL1 rs17047200 cytokine showed a higher incidence of the disease in comparison to the control group, with *p* = 0.012.

With regard to *DDR1* rs4618569, 66 (46.8%) patients were AA, 56 (39.7%) were AG and 19 (13.5%) were GG. With regard to the control group, 62 (62%) of them were AA, 24 (24%) were AG and 14 (14%) were GG (Table 2). From these data, it appears that the AA genotype of the rs4618569 variant of the *DDR1* gene had a higher incidence of COVID-19 compared to the healthy control group (*p* = 0.026) (Table 2).

We applied the Hardy–Weinberg equation to determine whether that the frequency of each genotype obtained agrees with expected values, as calculated from allele frequencies. Its value was 0.45 for cases and 0.83 for controls with regard to the first SNP, and 0.034 for cases and 0.035 for controls with regard to the *TLL1* rs17047200SNP and 0.25 for cases and 0.00039 for controls with regard to the *DDR1* rs4618569SNP. Haplotype analysis was done with a global haplotype association, *p*-value = 0.56 (Table 2).

Of note, there were no statistically significant differences between the different genotypes regarding the demographic and laboratory features. With regard to the IFN-λ rs12979860SNP, the TC genotype was associated with the presence of comorbidities and an increased mortality rate compared to the other genotypes. With regard to the TLL1 rs17047200 SNP, the AA genotype was associated with an increased risk for comorbidities and an increased risk for ventilation, and a decreased total leucocytic count was associated with disease severity with increased mortality rate; a significant difference was found between the different genotypes of SNP 2 (rs17047200) and CRP and the severity of the disease (*p* = 0.01, *p* = 0.02, respectively). With regard to the *DDR1* rs4618569SNP, the AG genotype was associated with an increased risk of comorbidities and an increased risk of ventilation, high CRP, Ferritin, D-dimer, and disease severity with increased mortality rate (Appendix A). 

Interestingly, regarding the *IFN-λ* rs12979860 SNP (rs12979860), 27 (45.8%) cases of the TC genotypes were classified as a severe disease compared to 22 (34.9%) and 10 (52.6%) cases in the CC and TT genotype, respectively, with *p* = 0.283. With regard to the *TLL1* rs17047200 SNP, in 36 (38.3%) cases of the AA genotype, the disease was severe in comparison to 16 (43.2%) cases in the AT genotype and 7 (70%) cases in the TT genotype, with *p* = 0.152. With regard to the *DDR1* rs4618569SNP, in 34 (60.7%) cases of the AG genotype, the disease was severe in comparison to 20 (30.3%) cases in the AA genotype and 5 (26.3%) cases in the GG genotype, with *p* < 0.01* (Appendix A). 

### 3.3. Genotypes and Outcome of the Disease

There were significant differences between the *DDR1* rs4618569SNP and the outcome of the disease, with the highest mortality in the AG genotype: 29 (60.4%) in comparison to 16 (33.3%) and 3 (6.2%) in the AA and GG genotypes, respectively (*p* = 0.007*), suggesting that the A allele is associated with a poor outcome in the disease. With regard to the *IFN-λ* rs12979860 SNP, the poor outcome is associated with the C allele and with the A allele in the TLL1 rs17047200 SNP (Table 3).

## 4. Discussion

The COVID-19 crisis represents a worldwide health problem. By now, the global number of confirmed cases of COVID-19 reached 108.2 million cases with a mortality of 2.3 million cases. In Egypt, they reached 177,543 confirmed cases and 10,298 deaths [22]. Developing countries face a critical financial problem with their limited resources that highlights the urgent need for the identification of high-risk patients who develop complications to receive timely and effective therapeutic regimens [23]. A study by Ellinghaus et al. has tested for association between >8 million single nucleotide polymorphisms (SNPs) and the development of respiratory failure in COVID-19 patients [24]. Therefore, understanding crucial players in the cytokine storm-related mortality in COVİD-19 is very important [25]. The work aimed to study the association of rs12979860, rs17047200 cytokines polymorphism and DDR1 gene variant rs4618569 with the progression and outcome in SARS-CoV-2 patients.

TaqMan based genotyping revealed that the frequency of the *IFN-λ* 3 SNP (rs12979860) showed that the CC genotype is more expressed in COVID-19 patients versus healthy controls (*p* = 0.011). Nineteen (13.5%) patients were TT, 59 (41.8%) were TC and 63 (44.7%) were CC versus 12 (12%), 44 (44%) and 44 (44%) in the healthy control group, respectively. These data suggest that persons who had the C allele (TC and CC) are at a higher risk to develop COVID-19. While the AA genotype of the *TLL1* rs17047200 had higher expression in patients versus healthy controls, with *p* = 0.012. Ninety-four (66.7%) patients were AA, 37 (26.2%) were AT and 10 (7.1%) were TT versus 66 (66%), 26 (26%) and 8 (8%) in the control group, respectively. These results demonstrate that the persons carrying the A allele (AA and AT) are at higher risk of the disease. Concerning the AA genotype of the *DDR1* rs4618569 variant, it is more likely to be found in COVID-19 patients versus healthy controls, with *p* = 5.422. Sixty-six (46.8%) patients were AA, 56 (39.7%) were AG and 19 (13.5%) were GG versus 62 (62%), 24 (24%) and 14 (14%) in the healthy control group, respectively. These results show that persons with the A allele, either AA or AG, are more susceptible to COVID-19.

In this COVID-19 global pandemic, we aim to identify host genomic factors that increase susceptibility and the complications of such a viral infection, and to translate these results in a timely manner to enhance patient care [26].

In our study, all patients showed higher levels of TLC, CRP, D-dimer, serum ferritin, LDH and lymphopenia, in agreement with Zhu et al. who found higher levels of these parameters in patients with COVID-19 infection [27].

Upon comparison of COVID-19 infection with different genotypes with regard to the disease severity and prognosis: The TC genotype of the *IFN-λ* 3 SNP (rs12979860), the AA genotype of TLL1 rs17047200 and the AG genotype of DDR1 rs4618569 variant are associated with more severe symptoms and lower favorable outcomes compared to the other genotypes. In the TC and CC genotypes of the *IFN-λ* 3 SNP (rs12979860), the disease was severe in 27 (45.8%) and 22 (34.9%) of cases, respectively, in comparison to 10 (52.6%) cases in the TT genotype, *p* = 0.283. Mechanical ventilation was used in 24 cases of the TC genotype and 21 cases of the CC genotype compared to 9 cases in the TT genotype. In the AA genotype of *TLL1* rs17047200, COVID-19 was severe in 36 (38.3%) cases compared to 16 (43.2%) cases in the TA genotype and 7 (70%) cases in the TT genotype, with *p* = 0.152. Mechanical ventilation was required in 31 cases of the AA genotype compared to 16 cases in the TA genotype and 7 cases in the TT genotype (*p* = 0.056).

In the AG genotype of the *DDR1* rs4618569 variant, the disease was severe in 34 (60.7%) cases compared to 20 (30.3%) cases in the AA genotype and 5 (26.3%) cases in the GG genotype (*p* = 0.001*). Mechanical ventilation was required in 32 cases of the AG genotype in comparison to 17 cases in the AA genotype and 5 cases in the GG genotype (*p* = 0.001*). These data revealed that the C allele of the IFN-λ 3 SNP (rs12979860), the A allele of *TLL1* rs17047200 and the A allele of the DDR1 rs4618569 variant are associated with more aggressive disease, and this may be correlated to changes in the product’s level of these loci in blood.

The rs12979860 SNP encodes the IFN-λ family of cytokines [28], which shared in the regulation of the host immune response against viral infections [29]. Many associations have been found between interferon lambda 3 rs12979860 polymorphism and various clinical outcomes as a control of the hepatitis C virus infection [30], myeloproliferative neoplasms, dengue virus in children and systemic lupus erythematosus [31].

In a GWAS, a research group declared the association between the *TLL1* rs17047200 SNP and hepatocellular carcinoma pathogenesis and found elevated *TLL1/TLL1* mRNA in animal models of liver injury and human liver tissues with fibrosis, in comparison with controls [32]. A previous study highlighted the association between rs4618569 of the *DDR1* and vitiligo development [33]; the DDR1 gene seems to be an important player in immune responses, which depend on the effective migration of activated leukocytes into infectious or inflammatory tissue sites [34].

Of interest, regarding the prognosis and survival of COVID-19, the TC, AA and AG genotypes of the SNPs *IFN-λ* 3 SNP rs12979860, *TLL1* rs17047200 and *DDR1* rs4618569 variant, respectively, showed poor prognosis, where 23, 29 and 29 cases died, respectively, in comparison to lower deaths in the other genotypes. Our data showed an increased risk of death from COVID-19 in patients with advanced age, in agreement with Wang et al. who found that elderly males showed poor prognosis and an increased risk of COVID-19 severity [35], which could be explained by a decline in the immune function with aging and a decrease in the production of CD3 + T cells, and B lymphocytes, along with elevated regulatory T cells [36,37]. Another study confirmed that the poor outcome in elderly patients with SARS-CoV-2 was due to immunosenescence [38].

We suggest larger multicenter studies to explore the validity of these polymorphisms in COVID-19 patients. The study may be limited by the patients and by being a single- center study.

## 5. Conclusions

The C and A alleles of SNPs IFN-λ rs12979860, TLL1 rs17047200 correlate with the severity of COVID-19, where the AG genotype of the DDR1 rs4618569 variant are associated with severe COVID-19 cases and poor outcomes. These data suggest that innate immunity is closely linked with the outcome of SARS-CoV-2 infection.

## Figures and Tables

**Table 1 genes-12-00830-t001:** Clinicopathological and laboratory data between the investigated groups.

	Cases *n* = 141			Control *n* = 100			Test of Significance
Sex							
Male 137 (56.8%)	85 (60.3%)			52 (52%)			χ^2^= 1.637
Female 104 (43.2%)	56 (39.7%)			48 (48%)			*p* = 0.201
Comorbidities							
+ve 99 (41.1%)	81 (57.4%)			18 (18%)			χ^2^= 37.613
−ve 142 (58.9%)	60 (42.6%)			82 (82%)			*p* = 0.000 *
Severity				NA			NA
Mild	82 (58.2%)						
Severe	11 (7.8%)						
Critical	48 (34%)						
Ventilation				NA			NA
+ve	54 (38.3%)						
−ve	87 (61.7%)						
Outcome				NA			NA
Recovery	63 (44.7%)						
Recurrence	30 (21.3%)						
Death	48 (34%)						
	**Median**	**Mean**	**±SD**	**Median**	**Mean**	**±SD**	**Test of significance**
Age/years	38	37.42	20.51	33	34.27	21.62	T = 1.149
							*p* = 0.252
Total leucocytic count (TLC) (thousands/cmm)	9.3	17.41	29.54	7	9.36	11.59	T = 2.58
							*p* = <0.01 *
Lymphocytes (×10^9^/L)	1.06	1.16	0.52	3.3	3.24	0.86	T = −23.23
							*p* = < 0.01 *
Hemoglobin (g/dL)	12.8	12.09	2.07	12.3	12.21	2.46	T =0.390
							*p* =0.697
Platelets (PLTs) (thousands/cmm)	261	260.31	77.89	222	244.52	99.78	T = 1.37
							*p* = 0.169
C-reactive protein (mg/L)	39.5	70.60	78.83	0.3	0.88	1.34	T = 8.838
							*p* = < 0.01 *
Serum Lactate dehydrogenase (LDH) (U/L)	271	376.32	251.20	147	166.52	50.39	T = 8.231
							*p* = < 0.01 *
Ferritin (ng/mL)	165	361.12	418.87	31.5	42.27	31.97	T = 7.59
							*p* < 0.01 *
D-dimer (mg/L)	5	132.05	430.95	0.10	0.12	0.10	T = 3.06
							*p* = < 0.01 *

t: Student *t*-test; χ^2^: Chi-Square test; TLC: Total Leucocytic Count; Comorbidities include diabetes mellitus, hypertension, asthma and combined comorbidities; * statistically significant.

**Table 2 genes-12-00830-t002:** Genotype frequencies between the studied groups.

Genotypes	Cases (141) *n* (%)	Control (100) *n* (%)	Test of Significance χ^2^	Odd’s Ratio (95% CI)
	IFN-λ rs12979860	Genotypes		
TT *n* = 31 (13%)	19 (13.5%)	12 (12%)	0.242 (0.702)	0.876 (0.404–1.897)
TC *n* = 103 (43%)	59 (41.8%)	44 (44%)	0.111 (0.792)	1.092 (0.651–1.832)
CC *n* = 107 (44%)	63 (44.7%)	44 (44%)	0.011 * (1.00)	0.973 (0.581–1.630)
Allele				
T *n* = 165 (34%)	97 (34%)	68 (34%)		
C *n* = 317 (66%)	185 (66%)	132 (66%)		
Hardy-Weinberg Equilibrium	0.45	0.83		
TLL1 rs17047200 Genotypes				
AA *n* = 160 (66%)	94 (66.7%)	66 (66%)	0.012 * (1.00)	0.971 (0.564–1.669)
AT *n* = 63 (26%)	37 (26.2%)	26 (26%)	0.002 * (1.00)	0.988 (0.551–1.770)
TT *n* = 18 (7%)	10 (7.1%)	8 (8%)	0.070 (0.808)	1.139 (0.433–2.996)
Allele				
A *n* = 383 (79%)	225 (80%)	158 (79%)		
T *n* = 99 (21%)	57 (20%)	42 (21%)		
Hardy–Weinberg Equilibrium	0.034	0.035		
DDR1 rs4618569 Genotypes				
AA *n* = 128 (80%)	66 (46.8%)	62 (62%)	5.422 (0.026)	1.854 (1.100–3.125)
AG *n* = 80 (33%)	56 (39.7%)	24 (24%)	6.517 (0.012)	0.479 (0.271–0.847)
GG *n* = 33 (14%)	19 (13.5%)	14 (14%)	0.014 * (1.00)	1.045 (0.497–2.198)
Allele				
A 336 (70%)	188 (67%)	148 (74%)		
G 146 (30%)	94 (33%)	52 (26%)		
Hardy–Weinberg Equilibrium	0.25	0.00039 *		

χ^2^: Chi-Square test; * statistically significant.

**Table 3 genes-12-00830-t003:** Risk factors of death among studied cases (COVID-19).

	Recovery *n* = 63	Recurrence *n* = 30	Death *n* = 48	Test of Significance
Age/years	51.08	74.33	95.06	KWχ^2^ = 31.846
				*p* = 0.000 *
Sex				X_2_ = 1.986 *p* = 0.370
Male 85 (60.3%)	42 (66.7%)	17 (56.7%)	26 (54.2%)	
Female 56 (39.7%)	21 (33.3%)	13 (43.3%)	22 (45.8%)	
Comorbidities				X_2_ = 14.396 *p* = 0.001 *
+ve 81 (57.4%)	26 (41.3%)	18 (60%)	37 (77.1%)	
−ve 60 (42.6%)	37 (58.7%)	12 (40%)	11 (22.8%)	
Ventilation				X_2_ = 104.381 *p* = 0.000 *
+ve 32 (22.7%)	2 (3.2%)	6 (20%)	46 (95.8%)	
−ve 109 (77.3%)	61 (96.8%)	24 (80%)	2 (4.2%)	
Severity				X_2_ = 105.151 *p* = 0.000 *
Mild (*n* = 82)	60 (95.2%)	22 (73.3%)	0 (0%)	
Severe(*n* = 11)	3 (27.2%)	8 (72.7%)	0 (0%)	
Critical(*n* = 48)	0 (0%)	0 (0%)	48 (100%)	
Haemoglobin (gm/dl)	69.64	104.10	52.09	KWχ^2^ = 30.226
				*p* = 0.000 *
TLC (thousands/cmm)	62.85	63.42	86.44	KWχ2 = 10.441
				*p* = 0.005 *
PLTs (thousands/cmm)	71.06	73.98	69.06	KWχ^2^ = 0.269
				*p* = 0.874
Lymphocytes (x10ˆ9/L)	79.40	88.22	49.22	KWχ^2^ = 21.811
				*p* = 0.000 *
Ferritin (ng/mL)	62.94	64.42	85.70	KWχ^2^ = 9.472
				*p* = 0.009 *
C-reactive protein (mg/L)	63.90	70.47	80.66	KWχ^2^ = 4.604
				*p* = 0.100
Serum LDH (U/L)	71.68	91.0	57.60	KWχ^2^ = 12.410
				*p* = 0.002 *
D-dimer(mg/L)	48.44	69.42	101.60	KWχ^2^ = 46.883
				*p* = 0.000 *
*SNP 1 (rs12979860)*				X_2_ = 5.590 *p* = 0.232
TT 19 (13.5%)	5 (7.9%)	6 (20%)	8 (16.7%)	
TC 59 (41.8%)	27 (42.9%)	9 (30%)	23 (47.9%)	
CC 63 (44.7%)	31 (49.2%)	15 (50%)	17 (35.4%)	
*SNP 2 (rs17047200)*				X_2_ = 3.801 *p* = 0.434
AA 94 (66.7%)	47 (74.6%)	18 (60%)	29 (60.4%)	
AT 37 (26.2%)	13 (20.6%)	10 (33.3%)	14 (29.2%)	
TT 10 (7.1%)	3 (4.8%)	2 (6.7%)	5 (10.4%)	
*SNP 3 (rs4618569)*				X_2_ = 14.193 *p* = 0.007 *
AA 66 (46.8)	33 (52.4%)	17 (56.7%)	16 (33.3%)	
AG 56 (39.7)	20 (31.7%)	7 (23.3%)	29 (60.4%)	
GG 19 (13.5)	10 (15.9%)	6 (20%)	3 (6.2%)	

## Data Availability

The data presented are available on request by the corresponding authors.

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
