# Peer review of "Association between Interferon-Lambda-3 rs12979860, TLL1 rs17047200 and DDR1 rs4618569 Variant Polymorphisms with the Course and Outcome of SARS-CoV-2 Patients"

_genes, 2021, doi:10.3390/genes12060830_

Round 1

Reviewer 1 Report

The author has thoroughly worked to understand the correlation of selected SNPs with COVID19 symptoms but there are few points that needs to be addressed.

  1. In Line 35-36, “People who carry C and A alleles of SNPs IFN-λ rs12979860, TLL1 rs17047200, AG genotype of DDR1 rs 4618569 variant are correlated with Covid-19 poor outcome”, is C and A allele is for both rs12979860 and rs17047200 or only for rs12979860? Please reframe this sentence.
  2. Could the authors please put multiple references for the line “Retrieved from literature and was found to be connected with the host viral immune defense. “Line 59.
  3. Could the authors put related Gene Names/rsIDs in “Table 2. Genotype frequencies between the studied groups”
  4. It would be better to please elaborate the statistical approach used to compare the values in table 2 as the % between cases and control are very close in row4 yet the significance is very good.
  5. Please remove double full-stop at line 173.
  6. It would be better to write rsIDs rather than 1st /2nd/3rd SNP for example in line 200.
  7. Please change “NOW” to “now” in discussion paragraph line 207.
  8. In line 268, “previous study highlighted the”, please change small p to P capital.
  9. Please change all rs IDs as rs[12345678] rather than Rs[12345678].
  10. In line 34 remove the full stop after bracket in “p=.007*).s”.
  11. Please give reasons to why has authors considered only these three polymorphisms in these three genes? Please specify the filtering that has been applied here. Are there any other known polymorphisms in these genes as well?
  12. In line 88, please change QRT-PCR instead of qRT-PCR.
  13. In line 181, Author has mentioned 1st SNP TC genotype has high comorbidities, looking at the data it seems like TT genotype has highest comorbidities as 73% is in TT, while only 52.5% in TC. Please explain.
  14. It is very surprising that authors have used less controls and more COVID19 cases, could you please elaborate the reason for this?

Author Response

Reviewer 1

(x) English language and style are fine/minor spell check required

As recommended by the reviewer, English and grammar have been checked all through MS

Authors response

  1. In Line 35-36, “People who carry C and A alleles of SNPs IFN-λ rs12979860, TLL1 rs17047200, AG genotype of DDR1 rs 4618569 variant are correlated with Covid-19 poor outcome”, is C and A allele is for both rs12979860 and rs17047200 or only for rs12979860? Please reframe this sentence.
  • AS suggested by the reviewer , this paragraph has been rephrases as follow
  • People who carry C and A alleles of SNPs IFN-λ rs12979860 & TLL1 rs17047200 respectively, AG genotype of DDR1 rs4618569 variant are correlated with Covid-19 poor outcome.
  1. Could the authors please put multiple references for the line “Retrieved from literature and was found to be connected with the host viral immune defense. “Line 59.
  • As suggested by the reviewer, refrences have been added to required paragraph
  • [1] Abdelwahab, S.F., Zakaria, Z., Sobhy, M. et al. Differential distribution of IL28B.rs12979860 single-nucleotide polymorphism among Egyptian healthcare workers with and without a hepatitis C virus-specific cellular immune response. Arch Virol 160, 1741–1750 (2015).
  • [1] Mehta, M., Hetta, H.F., Abdel-hameed, E.A. et al. Association between IL28B rs12979860 single nucleotide polymorphism and the frequency of colonic Tregin chronically HCV-infected patients. Arch Virol 161, 3161–3169 (2016). 
  • [1] Wróblewska, A., Bernat, A., Woziwodzka, A. et al. Interferon lambda polymorphisms associate with body iron indices and hepatic expression of interferon-responsive long non-coding RNA in chronic hepatitis C. Clin Exp Med 17, 225–232 (2017).

  1. Could the authors put related Gene Names/rsIDs in “Table 2. Genotype frequencies between the studied groups”
  • As suggested by the reviewer, Gene Names/rsIDs have been added to table 2
  1. It would be better to please elaborate the statistical approach used to compare the values in table 2 as the % between cases and control are very close in row4 yet the significance is very good.
  • As suggested by the reviewer, we used Chi square test as it is the only valid test to compare between qualitative variables (Genotype frequency between cases) and added clearly in the results. Taking into consideration, that Chi-Square value is calculated, it is extremely sensitive to sample size and distribution within the cells
  1. Please remove double full-stop at line 173.

As  suggested by the reviewer, double full-stop at line 173 has been removed

  1. It would be better to write rsIDs rather than 1st /2nd/3rd SNP for example in line 200.

As  suggested by the reviewer, rsIDs habe been added in line 200

  1. Please change “NOW” to “now” in discussion paragraph line 207.
  • As suggested by the reviewer, NOW”  has been changed to “now in paragraph line 207
  1. In line 268, “previous study highlighted the”, please change small p to P capital.
  • As suggested by the reviewer, small p  ah been changed to P capital
  1. Please change all rs IDs as rs[12345678] rather than Rs[12345678].
  • As suggested by the reviewer, all rs IDs as rs[12345678] has been modified.
  • In line 34 remove the full stop after bracket in “p=.007*).s”.
  • As suggested by the reviewer, full stop after bracket in “p=.007*).s”.has been deleted

  • Please give reasons to why has authors considered only these three polymorphisms in these three genes? Please specify the filtering that has been applied here. Are there any other known polymorphisms in these genes as well?
  • three polymorphisms were chosen based on linkage to COVID infection , host immune response, cytokine strom signaling with high ranking score uaing more than one database to decrease false discovery rate. Further validation has been done to verify the correlatiob between chosen SNPS and COVID 19 host immune response using gene ontology and Kegg map. Lastly we have verified the expression in different tissues using gene card database .
  • yes there are many other polymorphisms for these genes, we focused only on novel genes linked to COVID 19 and cytokine response
  • In line 88, please change QRT-PCR instead of qRT-PCR.
  • As suggested by the reviewer, qRT-PCR has been changed to  QRT-PCR
  • In line 181, Author has mentioned 1st SNP TC genotype has high comorbidities, looking at the data it seems like TT genotype has highest comorbidities as 73% is in TT, while only 52.5% in TC. Please explain.
  • Thanks to your comment, the percentage are made within SNP groups not within comorbidities.
  • It is very surprising that authors have used less controls and more COVID19 cases, could you please elaborate the reason for this?
  • Thanks to your valuable comments, we have initially enrolled 150 controls but they are excluded due to unfulfilling inclusion criteria, presence of other chronic diseases, Poor RNA quality after RNA extraction, Poor housekeeping gene expression. So finally we selected 100 healthy control fulfilling all inclusion criteria with good RNA quality

Reviewer 2 Report

The article is original, but it must be totally reread concerning the formatting (points, spaces coma, brackets….

The abstract should be more consistent

Many formatting corrections should be made some remarks are underlined in yellow in the document attached.

Some examples only in abstract: line 21/22, line 27/28/29/32/34/38/39 –punctuation, spaces, points, coma –

Homogenization and standardization of the references

All Values Below 0 – Zero – Must Show Zero

Author Response

Second reviewer

pen Review

English language and style

( ) Extensive editing of English language and style required
(x) Moderate English changes required
( ) English language and style are fine/minor spell check required
( ) I don't feel qualified to judge about the English language and style

Yes

Can be improved

Must be improved

Not applicable

Does the introduction provide sufficient background and include all relevant references?

(x)

( )

( )

( )

Is the research design appropriate?

(x)

( )

( )

( )

Are the methods adequately described?

(x)

( )

( )

( )

Are the results clearly presented?

( )

(x)

( )

( )

Are the conclusions supported by the results?

Moderate English changes required

·       As suggested by reviewer, English revision, grammar and punctuation has been revised all through MS

( )

(x)

( )

( )

Comments and Suggestions for Authors

The article is original, but it must be totally reread concerning the formatting (points, spaces coma, brackets….

  • As suggested by the reviewer, formatting has been modified

The abstract should be more consistent

  • As suggested by the reviewer, The abstract has been modified to be more consistent

Many formatting corrections should be made some remarks are underlined in yellow in the document attached.

Some examples only in abstract: line 21/22, line 27/28/29/32/34/38/39 –punctuation, spaces, points, coma –

  • As suggested by the reviewer, line 21/22, line 27/28/29/32/34/38/39 –punctuation, spaces, points, coma – and all through MS according to your valuable comments have been modified

Homogenization and standardization of the references

  • As suggested by the reviewer, homogenization and standardization of the references have been done

All Values Below 0 – Zero – Must Show Zero

  • As suggested by the reviewer, we showed Zero

Second reviewer

pen Review

English language and style

( ) Extensive editing of English language and style required
(x) Moderate English changes required
( ) English language and style are fine/minor spell check required
( ) I don't feel qualified to judge about the English language and style

Yes

Can be improved

Must be improved

Not applicable

Does the introduction provide sufficient background and include all relevant references?

(x)

( )

( )

( )

Is the research design appropriate?

(x)

( )

( )

( )

Are the methods adequately described?

(x)

( )

( )

( )

Are the results clearly presented?

( )

(x)

( )

( )

Are the conclusions supported by the results?

Moderate English changes required

·       As suggested by reviewer, English revision, grammar and punctuation has been revised all through MS

( )

(x)

( )

( )

Comments and Suggestions for Authors

The article is original, but it must be totally reread concerning the formatting (points, spaces coma, brackets….

  • As suggested by the reviewer, formatting has been modified

The abstract should be more consistent

  • As suggested by the reviewer, The abstract has been modified to be more consistent

Many formatting corrections should be made some remarks are underlined in yellow in the document attached.

Some examples only in abstract: line 21/22, line 27/28/29/32/34/38/39 –punctuation, spaces, points, coma –

  • As suggested by the reviewer, line 21/22, line 27/28/29/32/34/38/39 –punctuation, spaces, points, coma – and all through MS according to your valuable comments have been modified

Homogenization and standardization of the references

  • As suggested by the reviewer, homogenization and standardization of the references have been done

All Values Below 0 – Zero – Must Show Zero

  • As suggested by the reviewer, we showed Zero

 c
